# SafeKV: Safe KV-Cache Sharing in LLM Serving

kexin Chu , Zixu Shen , Dawei Xiang , and Wei Zhang*

School of Computing, University of Connecticut
{kexin.chu, qzt24001, ieb24002, wei.13.zhang}@uconn.edu

*Abstract*—**Global KV cache sharing significantly improves the efficiency of LLM inference but introduces substantial privacy risks, while strict per-user cache isolation protects user data at the cost of reduced performance—adding 8–38.9% overhead in time-to-first-token (TTFT) on LLaMA2-70B in our experiments. To bridge this gap, we present SafeKV, a privacy-aware KV cache management system that enables selective sharing of non-sensitive cache entries while isolating sensitive ones in private caches. SafeKV integrates ChunkGuard, a lightweight, real-time detector that classifies sensitive content at the chunk level, with a decoupled cache architecture consisting of a batched Cache Search Engine, Allocator, Monitor, and Evictor. This design supports constant-time batched prefix lookups and enforces fine-grained privacy policies with minimal overhead. By combining privacy-preserving inference with high cache reuse efficiency, SafeKV restores the benefits of global sharing while providing strong runtime privacy guarantees.**

## I. INTRODUCTION

Large Language Models (LLMs) have rapidly emerged as a transformative force, underpinning a wide array of applications from conversational AI to complex reasoning engines. With their vast knowledge and adaptive capabilities, LLMs have led to significant advancements in natural language understanding and generation. To support the increasing demand for real-time inference, Key-Value (KV) caching techniques have become indispensable for accelerating LLM performance. By storing intermediate hidden states—the "keys" and "values" generated during attention computations—KV caching reduces redundant computation, particularly in sequential or similar prompts [26]. This efficiency gain is amplified through KV cache sharing across multiple requests. In particular, prompts with common prefixes—such as shared dialogue history or structured prompting patterns—enable substantial throughput improvements and latency reduction. Consequently, shared KV caches play a vital role in enhancing performance and optimizing resource utilization in large-scale, multi-user LLM environments. Empirical observations indicate that a large proportion of real-world prompts exhibit structural or prefix-level commonalities, offering significant potential for cache reuse and improved responsiveness [11], [20], [29].

Despite these performance benefits, KV cache sharing raises serious privacy and security concerns in shared or multi-tenant deployments. When KV caches are reused indiscriminately across users or sessions, sensitive information—such as user inputs, context embeddings, personalized prompts, or reasoning traces—may be unintentionally exposed. Attackers can exploit such vulnerabilities through prompt manipulation, cross-request cache timing, or targeted probing to reconstruct hidden states and extract private data. These risks have been highlighted by a growing body of literature identifying diverse attack vectors, including prompt reconstruction and cache side-channel exploitation [17], [18], [20], [28]. Although various studies have examined these threats, few have explored practical defenses [8], [15], [21]. A straightforward defense is to isolate KV caches on a per-user basis, which eliminates sharing-induced leakage but also discards the performance gains of cache reuse. Recent benchmarks demonstrate that such isolation leads to considerable memory overhead and increased inference latency [10], [16], [25]. Notably, detailed analysis shows that only a small fraction of KV cache entries contain sensitive, user-specific data, while the majority are non-sensitive and may be safely shared [4], [24]. This insight opens the door to more refined strategies that aim to preserve both privacy and efficiency.

In response to these challenges, we propose a novel safe KV cache sharing framework that strikes a practical balance between privacy protection and system performance. Our key insight is the introduction of a privacy-aware KV caching mechanism, which we call the Selective Sharing Cache. This mechanism works by identifying private information, which is then isolated in a per-user private cache, while non-private information is safely promoted to a shared cache that can be accessed by all users without jeopardizing privacy. This innovative Selective Sharing Cache architecture enables secure cache sharing while maintaining the efficiency of cache reuse, opening up a new paradigm for KV cache management in multi-user LLM services. However, this approach also presents several critical technical challenges that must be addressed: 1) How can we accurately and efficiently distinguish between private and non-private cache entries in real time, while minimizing computational overhead? 2) How can we efficiently manage the lifecycle of private and shared caches to ensure that cache reuse is maximized without compromising security? 3) How can we perform efficient cache lookups when a query may partially hit both private and shared caches, without introducing unacceptable latency or performance bottlenecks? 4) In cases where privacy detection is imperfect, how can we mitigate the risk of information leakage if private data mistakenly enters the shared cache?

Addressing these challenges requires rethinking the core design of KV cache systems. Rather than focusing primarily

---

\* Corresponding Author.

on attack prevention, we take a proactive and performance-conscious approach, aiming to provide a scalable and secure solution for cache sharing that preserves the critical performance benefits of KV cache reuse. Our framework selectively isolates sensitive data with minimal disruption to the system's overall performance, offering a robust and scalable solution for next-generation LLM services that ensures privacy protection while retaining high efficiency.

- We quantify how global KV-cache sharing risks user privacy and show that per-user isolation severely degrades LLM inference performance.
- We introduce SafeKV, which isolates sensitive entries in per-user private caches while promoting non-sensitive entries to a shared cache, striking a balance between privacy and reuse.
- We develop ChunkGuard, a lightweight, real-time, chunk-level model for accurate privacy inference.
- We design a cache mechanism with a unified Cache Search Engine, Allocator, Monitor, and Evictor to optimize batched prefix lookups and enable low-overhead isolation.

## II. BACKGROUND AND MOTIVATION

### A. LLM Inference

Large Language Models (LLMs) based on the Transformer [19] architecture rely on scaled dot-product attention to compute Query (Q), Key (K), and Value (V) embeddings that capture contextual dependencies among tokens [9], [12]. Inference is divided into two phases. During the prefill phase, the entire input prompt is tokenized, and each token's K/V embeddings are computed across all layers in one bulk operation—yielding the first output token in a single step. In the decoding phase, tokens are generated one at a time: for each new token, Q/K/V embeddings are computed and attended against all cached context embeddings, incurring only linear growth in cost per token. Without caching, both phases exhibit quadratic time complexity in sequence length, making long prompts and multi-turn dialogues prohibitively expensive [7].

### B. KV-Cache Sharing

To eliminate redundant computation, LLM serving systems store K/V embeddings in GPU memory as a KV cache, reducing per-token cost to O(n) rather than O(n²). Early implementations allocated large static buffers, leading to fragmentation. Modern frameworks instead employ dynamic cache-block management:

- vLLM's PagedAttention divides the KV cache into small blocks, each tagged with a hash of preceding tokens, last-access timestamps, and reference counts [10]. An LRU policy evicts the oldest blocks when memory is constrained; later requests compare their prefix hashes to existing blocks and reuse matching embeddings.
- SGLang's RadixAttention [27] uses a radix tree keyed by token prefixes. Incoming requests are scheduled via Longest Prefix Match (LPM)—those sharing the longest

prefix with cached entries are prioritized—minimizing eviction and maximizing hit rates.

Both designs achieve dramatic TTFT reductions (cache hits in microseconds versus misses in milliseconds), enabling real-time, high-throughput LLM services even under limited GPU memory.

### C. Timing Side-Channel Attack

Although shared KV-cache optimizations significantly reduce inference latency, they also create a purely software-based side channel. An adversary with only black-box access can exploit timing differences as follows: [17], [20], [28]:

- Probe the service with a candidate prefix and measure TTFT.
- Detect a cache hit (low latency) if the victim has already populated that prefix, or a miss (higher latency) otherwise.
- Iterate token by token—using an incremental search algorithm—to recover private or proprietary prompt content with high precision.

Semantic caching (e.g., GPTCache [3]) exhibits analogous leaks: semantically similar probes yield millisecond-level hits versus second-level misses, revealing sensitive attributes in user queries. These attacks require only black-box access and have been demonstrated against both open-source and commercial LLM services.

### D. Threat Models

We assume attacker and victim reside in separate security domains and share access to a common LLM inference framework that uses prefix caching for performance. The attacker, acting as a benign user, cannot read the victim's private inputs directly but may exploit shared caches to probe for information about those inputs.

The attacker carries out a timing side-channel by issuing carefully constructed prompts to the LLM and measuring response delays. Cache hits induced by the victim's prior requests shorten inference time; by correlating latencies with prompt prefixes, the attacker can deduce which prefixes the victim has queried and thereby expose sensitive data. We do not consider physical attacks or other side channels such as early termination. A defense is considered secure if it masks cache-access patterns such that latency remains invariant regardless of the victim's activity.

### E. Motivation & Challenges

A straightforward solution to mitigate privacy risks in KV cache sharing is per-user cache isolation, which ensures that sensitive data is never shared. However, this comes at a significant performance cost: it eliminates cache reuse, leads to redundant computations, increases memory overhead, and degrades inference latency [14], [23]. These drawbacks render full isolation impractical for real-time, large-scale LLM services.

To address this, we propose the *Selective KV Cache Sharing* framework, which distinguishes private from non-private cache

entries. Sensitive data is kept in private caches, while non-sensitive entries are safely shared, preserving both privacy and performance. From Table I, we observe that a considerable portion of datasets still exhibit a high degree of inter-session reuse, indicating that completely disabling sharing can significantly impact overall system efficiency.

Realizing this framework presents several key technical challenges across four dimensions:

- **Privacy Detection.** A key challenge in selective cache sharing is accurately and efficiently identifying private versus non-private cache entries in real time [28]. Which may contain both sensitive data (e.g., personalized prompts, user-specific embeddings, or private reasoning traces) and non-sensitive data (e.g., shared model weights or intermediate results), making reliable classification essential. This task is complicated by the dynamic nature of inference, where cache contents vary with the prompt, user context, and reasoning path. Detection must operate with minimal latency to preserve performance and scale to high-throughput environments with frequent cache updates and diverse usage patterns. Achieving the right trade-off between detection accuracy and computational efficiency is critical for practical deployment.
- **Cache Lifecycle Management.** After classifying entries, managing their lifecycle is essential to maximize performance and ensure privacy [13]. Sensitive data must remain in private caches, while non-sensitive data should be promoted to the shared cache for reuse. This involves storing, invalidating, and reusing cache entries efficiently. Fine-grained control is necessary: premature invalidation increases overhead, while improper isolation risks privacy breaches. The challenge lies in maintaining secure handling of private data while maximizing reuse of non-sensitive content [22].
- **Cache Lookup Optimization.** In multi-user systems, queries may partially match entries in both private and shared caches, making lookups complex and potentially slow [5]. Multiple lookups increase latency, which is problematic for high-throughput, real-time inference. To avoid bottlenecks, lookup operations must be streamlined and scalable. Designing an efficient lookup strategy that minimizes delay when checking multiple caches is critical for maintaining low response times.
- **Mitigation of False Detection.** Privacy classification is not always perfect, and false negatives—where sensitive data is misclassified as non-sensitive—pose a serious risk. The system must contain the impact of such leaks to prevent privacy violations. Mitigation techniques may include secondary checks, fallback mechanisms, or encryption for shared cache entries. A dynamic response protocol is needed to adjust sharing behavior in real time when false detections are identified, minimizing potential harm and improving robustness.

Deploying selective KV cache sharing in multi-tenant LLM services requires carefully addressing these challenges. Suc-

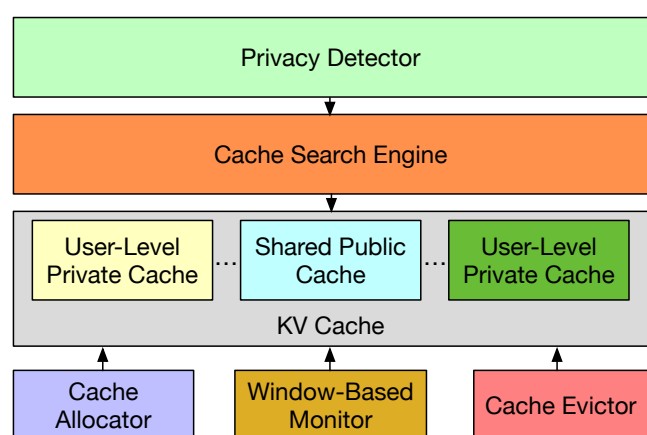

Fig. 1: The architecture overview.

cess depends on balancing privacy and performance: protecting sensitive data while enabling safe reuse of non-sensitive content. A scalable solution must minimize latency, preserve throughput, and ensure robust privacy guarantees, making it feasible for real-time deployment.

| Dataset | Intra-session Reuse (%) | Inter-Session Reuse (%) |
|---|---|---|
| ShareGPT V3 [1] | 7.06 | 25.49 |
| Multiturn Chat [2] | 31.47 | 9.45 |
| Prompt Multitasks [6] | 0.0 | 63.10 |

TABLE I: Intra-session and Inter-session KV-Cache reuse rates across different datasets.

## III. SYSTEM DESIGN

To address the privacy and performance challenges of KV-cache sharing in LLM serving, we propose SafeKV, a privacy-preserving KV-cache management framework illustrated in Figure 1. The processing pipeline is anchored by a Privacy Detector, which inspects input chunks to identify sensitive content. Depending on the result, the Cache Search Engine routes requests to either a User-Level Private Cache for isolated processing or a Shared Public Cache for maximizing reuse. These cache tiers operate within a unified KV Cache layer. To support efficient cache operations, SafeKVintegrates a Cache Allocator for dynamic memory provisioning, a Window-Based Monitor to detect abnormal high-frequency access patterns and mitigate misclassifications, and a Cache Evictor to maintain resource efficiency. Together, these components enable scalable, real-time, and privacy-aware cache sharing for LLM inference.

### A. ChunkGuard: Lightweight & Real-Time Privacy Detection

In the context of Selective KV Cache Sharing for Large Language Models (LLMs), a key challenge is designing a lightweight, real-time privacy detection mechanism that accurately differentiates private from non-private information without incurring significant overhead. As LLM services operate under high-throughput, low-latency, multi-user workloads,

traditional methods—such as full-query inspection or static filters—prove inadequate due to inefficiency and lack of flexibility. To address this, we propose **ChunkGuard**, a novel chunk-based, ML-driven privacy inference architecture designed for real-time deployment. ChunkGuard segments each prompt into fixed-size chunks and applies a lightweight, streaming-compatible transformer model to classify the privacy sensitivity of each chunk. This fine-grained approach enables early, localized detection, allowing sensitive segments to be routed into user-scoped KV caches while safely reusing non-sensitive segments in a shared cache. By combining heuristics, ML inference, and metadata tagging in a unified pipeline, ChunkGuard achieves both strong privacy protection and high-performance reuse—making it scalable and deployable for modern LLM-serving infrastructures.

At the core of ChunkGuard is its fixed-size chunking strategy. Instead of analyzing prompts holistically, ChunkGuard partitions them into uniform-length chunks prior to inference. This design simplifies preprocessing, enables parallel detection, and supports streaming workloads. Localized chunk-level detection reduces overhead and allows early identification of sensitive content, enabling timely routing decisions during real-time serving. Despite its simplicity, fixed-size segmentation maintains high accuracy when paired with a lightweight inference model, making it a practical solution for privacy-aware caching.

Each chunk is evaluated by a compact transformer-based privacy classifier, optimized for real-time inference. This component uses quantized or distilled models to reduce computational cost while preserving detection quality. Trained on a curated, privacy-annotated dataset containing sensitive entities (e.g., names, health data, personal identifiers), the model learns fine-grained privacy patterns and is well-suited for integration into high-throughput LLM pipelines.

To ensure robust classification, ChunkGuard employs adaptive confidence thresholding: a prompt is flagged as private if any chunk exceeds a defined sensitivity threshold. This conservative, safety-first strategy minimizes false negatives while avoiding excessive over-isolation, effectively balancing privacy protection and system utility. Based on the classification outcome, ChunkGuard dynamically routes private inputs to user-scoped KV caches to guarantee isolation, while directing non-sensitive inputs to a shared cache to maximize reuse. This integrated mechanism preserves the performance benefits of KV caching without compromising privacy guarantees.

A key advantage of ChunkGuard is its compatibility with streaming inference. By operating on partial input, it enables early classification and routing decisions before the full prompt is received. This is particularly beneficial in multi-turn dialogue or token-by-token generation, reducing end-to-end latency while maintaining privacy fidelity.

To further enhance protection, ChunkGuard's classifier is trained with differential privacy techniques, ensuring that sensitive training examples cannot be reverse-engineered. This adds a defense-in-depth layer, safeguarding user data during both inference and model development.

### B. Cache Search Engine: Batch Processing

In SafeKV's architecture, a hash-based KV cache design is employed to efficiently manage both privacy-aware and non-privacy KV cache entries. This design enables low-latency, batch-level prefix retrieval, which is essential for high-performance LLM inference while ensuring privacy isolation. To achieve this, each input prompt is partitioned into fixed-size chunks, allowing for the precise segmentation and storage of relevant data in the cache.

Each chunk is then mapped to a global hash table that connects the chunk's hash value to its associated attention-key (K-Cache) and attention-value (V-Cache) blocks. The key design feature of this system is how the hash for each chunk is calculated. Specifically, for chunk $i$, the hash is computed as follows: for non-privacy chunks, the hash is:

$$\text{hash}_i = \text{Hash}(\text{hash}_{i-1} \parallel \text{chunk}_i)$$

For privacy-sensitive chunks, the hash is computed by including the session identifier (session_id) to ensure that privacy isolation is maintained across different users' data:

$$\text{hash}_i = \text{Hash}(\text{hash}_{i-1} \parallel \text{chunk}_i \parallel \text{session\_id})$$

Here, $\text{hash}_{i-1}$ represents the hash of the previous chunk in the context, and the inclusion of the session identifier for privacy-sensitive chunks guarantees that the data remains user-specific, while non-sensitive data can be shared more freely.

Each entry in the global hash table contains several important elements: it includes a pointer to the associated Key-Cache (K-Cache) and Value-Cache (V-Cache) blocks, metadata fields such as last_acc_ts (timestamp of the last access), cache_hit_count_cur (current cache hit count), cache_hit_count_prev (previous cache hit count), and other relevant tracking information. Additionally, the entry contains a next-chunk pointer that ensures continuity in the caching of consecutive chunks, preserving context across chunks in the prompt.

To enhance search efficiency and reduce latency, SafeKVemploys batch processing for KV cache lookups. Instead of retrieving chunks individually, which would result in higher latency, batch processing allows the system to process multiple chunks at once. By grouping chunks that share similar prefixes, SafeKVcan perform efficient, parallel lookups, reducing the time it takes to fetch the required Key-Cache and Value-Cache blocks. This batch-based approach, combined with constant-time hash lookups, ensures that the system maintains high throughput and low latency, even under heavy request loads.

### C. Cache Allocator: Adaptive Provisioning

In SafeKV's architecture, the cache allocator goes beyond traditional static memory budgeting by intelligently adapting to the ever-changing privacy requirements of the system. This dynamic cache allocation mechanism ensures optimal resource utilization while maintaining robust privacy isolation and preserving high-performance throughput. By continuously

adjusting the partition sizes between the User-Level Private Cache and the Shared Public Cache, SafeKVis able to respond to fluctuating privacy demands without compromising system efficiency.

During LLM inference, key-value (KV) caches are generated and stored in high-bandwidth memory (HBM) for fast access. As each chunk of data is processed, the Cache Allocator begins by evaluating its privacy status based on the results from the Privacy Detector. Depending on this evaluation, the chunk is either assigned to the User-Level Private Cache—ensuring sensitive user data remains isolated—or to the Shared Public Cache, where non-sensitive data can be reused. Once the appropriate cache placement is determined, the allocator calculates the key- and value-cache addresses for the chunk and inserts a new entry into the global hash table. This entry contains not only the key-value pointers but also critical metadata, such as $last\_acc\_ts$ (timestamp of the last access), $cache\_hit\_count\_cur$ (cache hits in current time window), $cache\_hit\_count\_prev$ (cache hits in last time window), and other tracking information. To ensure consistency and continuity of cached data, the next-chunk pointer of the previous entry is also updated, preserving the integrity of cached chunks across sequential requests.

Rather than relying on fixed memory budgets for private and shared caches, SafeKVcontinuously monitors privacy detection statistics and adjusts cache partitioning in real-time. Leveraging metadata counters and privacy flags from incoming requests, the system dynamically allocates memory based on the current privacy needs. For instance, during periods of increased privacy-labeled requests—indicating a higher volume of user-specific data—SafeKVintelligently reallocates resources by shifting memory from the Shared Public Cache to the User-Level Private Cache, ensuring that privacy-sensitive data is properly isolated. This adaptive memory management strategy not only guarantees privacy compliance but also optimizes overall cache usage and system performance, adapting seamlessly to the dynamic needs of multi-user LLM services.

### D. Cache Eviction: Anomaly-Aware Privacy Protection

Although SafeKVincorporates a privacy classification mechanism to isolate sensitive content, it remains vulnerable to cache probing attacks, where adversaries infer private input prefixes by analyzing access patterns and response latencies. To counter this threat, SafeKVintegrates a lightweight, adaptive eviction mechanism driven by a novel anomaly detection module that monitors temporal access behavior in real time.

Each entry in the SafeKVhash table maintains metadata fields `hit_cur` and `hit_prev`, which record the number of accesses during the current and previous observation windows, respectively. Under normal usage, access frequency is expected to remain relatively stable. SafeKVflags potential anomalies when current accesses spike relative to the previous baseline. Specifically, a chunk is considered anomalous if:

$$\mathtt{hit\_cur} \geq 2 \times \mathtt{hit\_prev}$$

This conservative threshold is designed to detect abrupt changes without penalizing high-frequency but consistent traffic patterns.

Upon detecting an anomaly, SafeKVtriggers a targeted eviction protocol. Rather than removing the entire metadata record, which could disrupt downstream context, it selectively invalidates the associated attention-key and attention-value blocks in memory—effectively erasing the cached content while preserving structural integrity. This ensures that any potentially leaked data is promptly removed without disrupting unrelated cache entries.

This eviction strategy also serves as a secondary safeguard against misclassified content. While the classifier is trained to identify sensitive data, errors are inevitable. By incorporating runtime access patterns, SafeKVintroduces a dynamic, feedback-driven mechanism to detect and mitigate risks that static classification may miss.

Compared to conventional eviction policies (e.g., LRU or LFU) focused solely on performance, SafeKV's anomaly-aware strategy prioritizes privacy without significantly sacrificing throughput. It evicts entries only in the presence of suspicious behavior, preserving cache efficiency under normal conditions while strengthening defenses against timing-based inference attacks.

By combining static privacy detection with runtime anomaly monitoring, SafeKVprovides a dual-layer defense for shared caching infrastructure. This hybrid strategy—integrating privacy awareness, adaptive response, and lightweight mitigation—enhances the resilience of LLM caching systems under adversarial workloads.

## IV. EVALUATION

In this section, we evaluate SafeKVusing different LLM models such as Llama-2-13B, and Llama-2-70B-GPTQ. We utilize the following datasets: ShareGPT [1], Multiturn Chat [2], and Multitasks [6]. We implemented SafeKVbased on Sglang and deployed it on a system with 2 NVIDIA A6000-48GB GPUs.

### A. Isolated per User vs Global Sharing

We compare the time-to-first-token (TTFT) under isolation versus global-sharing for two models: LLaMA-2-13B and LLaMA-2-70B, across three datasets with different workload patterns (ShareGPT, multiturn and multitask). As shown in Figure 2(a) and (b). For LLaMA-2-13B, enabling isolation incurs only a modest TTFT overhead: roughly 3.2 – 2.3% in the ShareGPT and multiturn datasets, rising to about 8.9 % under the most diverse (multitask) workload. In contrast, LLaMA-2-70B exhibits a larger penalty—approximately 29.7 % for ShareGPT, 8.3 % for multiturn, and over 38.9 % for multitask—reflecting the increased cost of enforcing strict privacy isolation on a heavier model. These results show that while isolation can protect private data, it can impact inference performance, and the impact scales with model size and request diversity. For larger models, the impact of isolation-per-user is unacceptable.

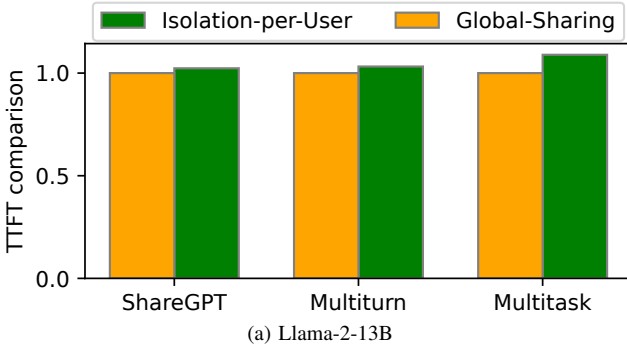

(a) Llama-2-13B

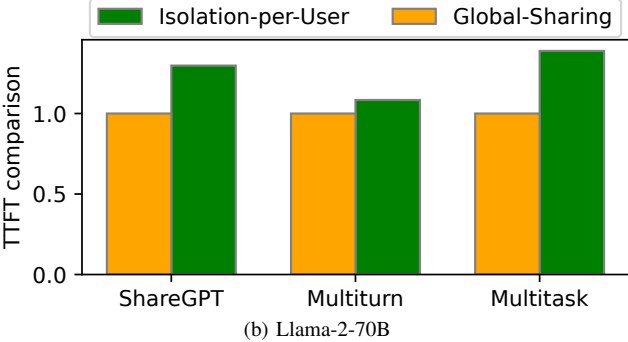

(b) Llama-2-70B

Fig. 2: Normalized performance of TTFT between global-sharing and the isolated-per-user cache reuse.

## V. CONCLUSION

We identify a fundamental privacy–performance tradeoff in KV-cache reuse: while per-user isolation safeguards sensitive data, it incurs significant overhead—ranging from 8% to 38.9% in time-to-first-token (TTFT) on LLaMA-2-70B. To address this, SafeKV selectively isolates sensitive cache entries while enabling the safe reuse of non-sensitive ones. This is achieved through ChunkGuard's real-time sensitivity detection and a decoupled architecture comprising a Cache Search Engine, Allocator, Monitor, and Evictor. Future work will complete the SafeKV prototype, aiming to restore near-global sharing performance while upholding strong privacy guarantees.

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
