# OpenReview forum: "SafeKV: Safe KV-Cache Sharing in LLM Serving"
_iscaconf.org/ISCA/2025/Workshop/MLArchSys — MLArchSys 2025 Oral_

### Official Review · Reviewer_7giX · 2025-05-12
**Review: SafeKV Cache Privacy Concept with Strong Potential, Empirical Validation Needed**

**Confidence:** 3
**Rating:** 6

**Detailed Feedback And Questions For Authors:**

Your work demonstrates significant originality in tackling the critical privacy-performance dilemma in LLM KV caches. The proposed system is presented with good clarity and showcases a high degree of conceptual quality. While the current draft would be further strengthened by empirical validation to fully substantiate its claims, the innovative ideas presented are of considerable significance and offer a valuable contribution to exploring new concepts in this space.

**Top Reasons To Accept The Paper:**

1.	The paper addresses a highly relevant and timely problem with a novel approach.
2.	The methods presented are innovative and could inspire new research directions.

**Top Reasons To Reject The Paper:**

1.	There is a lack of empirical performance evaluation of the SafeKV approach – makes the presentation seem possibly premature.

---

### Official Review · Reviewer_7DNQ · 2025-05-17
**Well-designed architecture for KV-cache privacy, but lacks experimental validation**

**Confidence:** 3
**Rating:** 4

**Detailed Feedback And Questions For Authors:**

#### Summary
This paper presents SafeKV, a well-motivated and elegantly designed KV-cache management system for LLM serving. It combines static privacy classification (via ChunkGuard) with runtime anomaly monitoring to protect against cache-based inference attacks, while preserving global cache reuse for non sensitive data. The proposed architecture is modular, conceptually sound, and tailored to the multi-tenant, high-throughput nature of modern LLM systems.

#### Suggestions for Additional Evaluation
- Quantitative analysis of misclassification risks for SafeKV:  premature invalidation, improper isolation, false detection
- A detailed latency breakdown for each stage (privacy classification, batch lookup, cache eviction, etc.)
- Concrete, experimental evidence showing whether SafeKV effectively masks latency signals, making attackers unable to infer private prefixes
- Deeper elaboration on the privacy classifier: its architecture, training process, and dataset composition
- Clarification on how SafeKV batches prefix chunks during lookup, and how chunk similarity is defined
- ablations for anomalous rate (r_a): hit_cur ≥ r_a × hit_prev. Prove that r_a=2 is the optimal design.


#### Questions.
- Does isolation introduce observable latency differences that attackers can exploit to identify sensitive inputs?
- Under the prefix hashing mechanism, is cache reuse entirely blocked if even a single sensitive chunk is detected in the prompt?
- In the cache eviction strategy:
  - Is there a risk that evictions triggered by an attacker's probing could negatively impact other users (victims) who are legitimately reusing the same chunk, leading to performance degradation such as increased cache misses or TTFT?
  - Was any “user diversity-aware” anomaly detection strategy considered to reduce collateral eviction of frequently reused cache chunks?

**Top Reasons To Accept The Paper:**

1. This paper clearly identifies the security challenges in shared KV cache systems and proposes a framework that isolates sensitive data while preserving performance through selective sharing.
2. It presents a novel and well-structured architectural design, including privacy classification, chunk-level granularity, and runtime anomaly detection for privacy-preserving and high-performance real-time LLM inference.
3. The proposed design represents a solid starting point for further work on secure and efficient KV cache management in multi-tenant LLM services.

**Top Reasons To Reject The Paper:**

1. The evaluation provides a good starting point, but further empirical validation is needed to fully demonstrate the system’s effectiveness. The only reported metric is time-to-first-token (TTFT), without supporting evidence for core components like ChunkGuard’s classification accuracy, cache reuse efficiency, or robustness against adversarial probing.
2. The large disparity in TTFT overhead between LLaMA-2-13B and 70B models (ranging from ~3% to nearly 39%) is left unexplained.
3. Some of the paper’s foundational assumptions and methods could benefit from more detailed empirical validation.—for example, the assertion that user-specific sensitive data constitutes only a small fraction of cache entries, and the design/accuracy of the privacy classifier and its training dataset are not detailed.
4. It would be helpful to clarify whether certain design decisions effectively mitigate timing-based leakage under realistic attack conditions. Some components may inadvertently reveal information through latency differences. Specific questions are raised below.

---

### Official Review · Reviewer_KLmR · 2025-05-18
**SafeKV: Safe KV-Cache Sharing in LLM Serving**

**Confidence:** 3
**Rating:** 5

**Detailed Feedback And Questions For Authors:**

1. How is the sensitivity threshold defined?
2. What are the overheads of dynamically partitioning user-level and shared caches? Can there be fragmentation? How are you handling it?
3. What are the overheads of continuously monitoring privacy detection statistics?
4. What are the overheads of the anomaly detection module?
5. Describe the legends in Figure 2. What are you comparing: full isolation vs full sharing? Where is SafeKV here?
6. Figure 2 shows the overall penalty. Can you show the breakdown?
7. What are the differences between the workload patterns? Can you correlate those differences with the different penalties you observe?

**Top Reasons To Accept The Paper:**

The authors propose a privacy-preserving KV Cache management framework that works on chunks of prompts to classify sensitive vs non-sensitive entries and puts them in their respective caches. The paper tries to solve a very timely problem in a way such that performance is the least impacted compared to fully isolating the KV Caches of all the users. The authors also explain the critical aspects that need to be handled in these scenarios, such as dynamic cache allocation and cache eviction policy prioritising privacy.

Overall, the paper addresses all the important aspects needed for privacy preserving, along with maintaining inference performance and maximising reuse.

**Top Reasons To Reject The Paper:**

The evaluation is incomplete. After having described many unique features, there is no evaluation to show the overhead of these features.

Specific questions below

---

### Official Review · Reviewer_tDJo · 2025-05-18
**Interesting problem, but a solution yet to be evaluated**

**Confidence:** 2
**Rating:** 3

**Detailed Feedback And Questions For Authors:**

In its current state, the paper focuses on motivating the problem and provides a sketch of their proposed solution. For future development (and perhaps even a brief mention if any very early numbers exist), providing data on SafeKV's actual performance (TTFT, throughput under varying sensitivity ratios) compared to global sharing and full isolation would be highly impactful. What are your anticipated performance characteristics?

Implement and thoroughly evaluate SafeKV (and in particular ChunkGuard). This must include end-to-end performance (latency, throughput) compared to baseline global sharing and per-user isolation under various workloads.
Provide a detailed description and evaluation of ChunkGuard: its architecture, training, inference overhead, and classification accuracy on realistic datasets.
Conduct a rigorous security analysis. Demonstrate, perhaps through simulated attacks or formal reasoning, how SafeKV mitigates the identified privacy risks, including timing side-channels.

As of now, the router proposed for private and shared caches presents security concerns as learning-based models can not provide formal guarantees. The main selling point being privacy, the authors should redirect their efforts into the isolation process rather than performance.

**Top Reasons To Accept The Paper:**

The paper addresses a critical and timely problem in online LLM serving: the trade-off between the performance benefits of KV cache sharing and the associated privacy risks.
The paper clearly articulates the technical challenges involved in such a system (privacy detection, cache lifecycle, lookup optimization, false detection mitigation) and proposes a plausible high-level architecture.

**Top Reasons To Reject The Paper:**

The authors propose a heuristic and ML-based classifier (ChunkGuard) to classify private and shared data. This is the central component of their architecture, as it is ultimately the router for private and shared cache entries.
By being a trained classifier, ChunkGuard can not formally guarantee privacy and therefore SafeKV can not pretend to provide strong isolation: false negatives can potentially expose private data in the shared cache, and if they exist, they could be exploited.
Instead, ChunkGuard could pretend to mitigate privacy concerns. However, there is no quantitative evaluation of ChunkGuard's effectiveness making it difficult to gauge the extent of its capabilities, furthermore the authors only vaguely describe the architecture, training process and datasets used for the classifier.
The anomaly detection sounds too simplistic, the heuristics are mentioned but not described.